# A Clinical and Genetic Evaluation of Cases with Folate Receptor α Gene Mutation: A Case Series from Türkiye

**DOI:** 10.3390/diagnostics15070892

**Published:** 2025-04-01

**Authors:** Abdurrahman Akgun, Ibrahim Tas

**Affiliations:** 1Division of Metabolism, Department of Pediatrics, Firat University School of Medicine, 23200 Elazig, Türkiye; 2Clinic of Pediatric Metabolic Diseases, Umraniye Training and Research Hospital, University of Health Sciences, 34766 Istanbul, Türkiye; ibrahimtas@hotmail.fr

**Keywords:** cerebral folate transporter deficiency, FOLR1, 5-methyltetrahydrofolate, folic acid, folinic acid

## Abstract

**Background/Objectives**: Cerebral folate transporter deficiency is characterized by pauses and regression in general development stages, with ataxia, choreoathetoid movements, and myoclonic epilepsy generally resistant to treatment. The aim of this study was to comprehensively evaluate cases followed up in two centres in Türkiye for a diagnosis of folate receptor-α deficiency. **Methods**: The study included nine cases from six different families. **Results**: The patients comprised 22.2% males and there was parental consanguinity in 88.9% of cases. The mean age at which complaints were first noticed was 3.7 years, and the age of definitive diagnosis was 10.4 years. The most frequently seen first complaints were febrile convulsions and attention deficit-hyperactivity-learning difficulties. The diagnosis most commonly made before the definitive diagnosis was epilepsy, and the first seizure occurred at a mean of 5.2 years. On cranial imaging, white matter involvement, cerebellar atrophy and cerebral atrophy were determined most often. Definitive diagnosis was established solely through clinical findings and genetic analysis. Three different variants in the *FOLR1* gene were determined. Treatment with folinic acid at a dose of 5.2 mg/kg/day of PO was started at the age of 9.8 years on average, and intravenous folinate was started at different doses. **Conclusions**: This study stands out as one of the largest case series in the literature and identifies a previously unreported novel variant. Our study suggests that FOLR1-related CFD should be considered in cases with febrile convulsions, developmental delay, ataxia, autism spectrum disorder, acquired microcephaly, and MRI findings of white matter involvement and cerebellar atrophy. Due to an asymptomatic early period, CFD diagnosis may be delayed, and treatment after symptom onset may be less effective. Incorporating *FOLR1* gene analysis into newborn screening programmes could facilitate early diagnosis and treatment. It is thought that the application of vagus nerve stimulation, in addition to folinic acid and anticonvulsant drug treatment, could be effective in seizure control.

## 1. Introduction

Cerebral folate transporter deficiency (CFD) is a disease with autosomal recessive transmission due to mutations in folate receptor 1 gene (*FOLR1*) (OMIM#613068), which encodes folate receptor alpha (FRα) [1]. Pauses and regression in general development stages, ataxia, choreoathetoid movements, and myoclonic epilepsy are generally resistant to treatment are seen in CFD [2].

The primary folate form in plasma is 5-methyltetrahydrofolate (5MTHF), which is a reduced folate form that can pass through the choroid plexus [3]. The cellular uptake of 5MTHF is provided by the proton-coupled folate transporter (PCFT), reduced folate carrier (RFC), and the two GPI-bound receptors, folate receptor alpha (FRα) and beta (FRβ) [4]. 5MTHF crosses the blood–brain barrier in the choroid plexus via the GPI-anchored membrane FRα and PCFT, facilitating its transport into the cerebrospinal fluid (CSF) [3,5]. FRα is generally found in the choroid plexus, lungs, thyroid, and in epithelial cells such as renal tubular cells, whereas FRβ is in mesenchymal stem cells such as red blood cells. Both are receptors with high affinity to folate [4,6].

Folates play a critical role in cellular single-carbon metabolism by acting as givers and receivers of single-carbon units in the synthesis of thymidine and purines and in the synthesis and catabolism of amino acids (serine, glycine, methionine, homocysteine, and histidine) [7].

The aim of this study was to comprehensively evaluate the clinical, laboratory, and genetic characteristics of cases diagnosed with folate receptor-α deficiency that were followed up in two different centres in Türkiye.

## 2. Materials and Methods

Cases diagnosed with folate receptor alpha deficiency were included in the study without age restrictions until January 2025 at the Pediatric Metabolism Clinic of Elazig Fırat University Hospital and the Pediatric Metabolism Clinic of the University of Health Sciences Istanbul Umraniye Training and Research Hospital. Cases that were not determined with a mutation in the *FOLR1* gene were not included in the study. The data were collected retrospectively from the patient files. A record was made for each patient for gender, diagnosis centre, age at diagnosis, current age, parental consanguinity, clinical findings, hemoglobin, MCV, folate, homocysteine, brain imaging findings, EEG and EMG results at the time of presentation, *FOLR1* gene analysis, treatments received, clinical responses during follow-up.

Data were analyzed statistically using the Statistical Package for the Social Sciences vn. 22 software (SPSS Inc., Chicago, IL, USA) with descriptive statistical methods and the Wilcoxon test.

## 3. Results

An evaluation was made for nine cases from six different families, including three cases from family number one, two cases from family number three, and one case from each of the other families. The cases comprised 22.2% males and 77.8% females. Parental consanguinity was present in 88.9% of the cases. Epilepsy, at the rate of 44.4%, was the most common diagnosis made during the diagnostic process. The other diagnoses are shown in Table 1. The first complaints were noticed at the mean age of 3.7 ± 1.6 years (range, 0.7–6 years). The mean age at the time of diagnosis was 10.4 ± 6.3 years (range, 4.6–23.5 years) and the current age was 12.2 ± 6.5 years (range, 6–26 years). The complaints for the cases in the first presentation are presented in Table 2.

The neurological and developmental steps during the diagnosis of the cases are shown in Figure 1. Among behavioural disorders, anxiety, attention deficit hyperactivity, and social withdrawal were each observed at a rate of 22.2% (*n* = 2). Autism spectrum disorder was present in one (11.1%) case.

When examined in respect of movement disorders, ataxia was present in 44.4% (*n* = 4) of cases, and spastic quadriplegia was present in 33.3% (*n* = 3). There was hypertonia in the extremities in 33.3% (*n* = 3) of cases and no significant tonus change in any of the other cases. Microcephaly was determined in 55.6% (*n* = 5) of cases. In the deep tendon reflex (DTR) examination, there was reduced DTR in 44.4% of cases, brisk in 11.1% of cases, and normal conditions observed in the remaining cases. Although EMG studies could not be performed in all cases, motor-dominant-generalized polyneuropathy with axonal damage was detected in 22.2% (*n* = 2) of cases, while generalized sensorimotor polyneuropathy with axonal damage was identified in 11.1% (*n* = 1) of cases. The EMG examination was evaluated as normal in 22.2% of cases. The age at which the first seizure was seen was a mean of 5.2 ± 3.2 years (range, 1.5–11 years). Many varied seizures were seen in each of the seven (77.8%) cases, and tonic seizures were seen in one case (11.1%). In one case determined with family screening only, there was no history of seizures. The EEG results were evaluated as abnormal in 77.8% (*n* = 7) of the cases. There was a history of ≥3 status epilepticus in 44.4% (*n* = 4) of cases and a history of 1 status epilepticus in 22.2% of cases (*n* = 2).

The full blood count and biochemical test results of the cases are shown in Table 3. As 5MTHF in CSF cannot be tested in many centres in Türkiye, including the two centres where the study was conducted, the CSF 5MTHF level was not examined.

As a result of the cranial magnetic resonance imaging (MRI), cerebral atrophy was determined in 44.4% (*n* = 4) of the cases, cerebellar atrophy was determined in 66.7% of cases, white matter involvement was observed in all the cases, white matter encephalomalacia was observed in 11.1% of cases (*n* = 1), corpus callosum atrophy was observed in 11.1% of cases (*n* = 1), diffuse thickening in the calvarium was observed in 11.1% of cases (*n* = 1), and basal ganglion calcification was observed in 11.1% of cases (*n* = 1). Brain MR spectroscopy could only be performed in four cases, and of these, a minimal increase was determined in the choline level and a minimal decrease was determined in the inositol level in only one case. In the follow-up period after treatment, brain MRI could be performed in six cases. No improvement or worsening was seen in follow-up brain MRI in 44.4% of cases (*n* = 4), partial recovery was determined in 11.1% of cases (*n* = 1), and full recovery was seen in 11.1% of cases (*n* = 1).

Definitive diagnosis could be made for the cases using genetic analysis. The diagnosis was made in 44.4% of cases (*n* = 4) with WES, in 33.3% of cases (*n* = 3) with CES, and in 22.2% of cases (*n* = 2) with *FOLR1* gene sequence analysis. Variants were determined to be homozygote in all the cases. Based on the frequency of occurrence, the identified variants in the *FOLR1* gene were c.610C>T (p.R204Ter) (77.8%), c.591C>A (p.Y197Ter) (11.1%), and c.466T>G (p.W156G) (11.1%). Only the c.466T>G (p.W156G) variant was a missense mutation, and the other two variants were nonsense mutations. While all variants were generally classified as pathogenic according to in silico predictors, in ClinVar, the c.610C>T (p.R204Ter) variant was reported as likely pathogenic, whereas no information was available for the other two variants. The c.610C>T (p.R204Ter) and c.466T>G (p.Trp156Gly) variants are previously reported variants, whereas the c.591C>A (p.Tyr197Ter) variant was determined to be a novel variant reported for the first time.

Calcium folinate treatment could be started for cases at the mean age of 9.8 ± 6.3 years (range, 4.6–23.5 years). Oral calcium folinate was administered at the dose of mean 5.2 ± 1.2 mg/kg/day (range, 3–7.5 mg). One case (11.1%) did not start IV treatment as clinical recovery was obtained with oral calcium folinate treatment. In the other cases, IV calcium folinate treatment was started once a week in 66.7% of cases (*n* = 6) and once a month in 11.1% of cases (*n* = 1). Case number 1 was admitted to the Intensive Care Unit (ICU) as the general condition was poor when diagnosis was made, and treatment was started with 100 mg calcium folinate IV daily. However, that patient was exitus after 3 days.

Five of the cases (55.6%) receiving IV treatment once a week were administered 100 mg of calcium folinate. Case number 4 was diagnosed at the age of 23 years. When the definitive diagnosis was made, the patient was dependent on a mechanical ventilator in the ICU because of status epilepticus and treatment was started with 200 mg/day of calcium folinate IV (for one week) until oral calcium folinate could be obtained. Within 1 week of starting the treatment, the seizures were brought under control, and the patient could be extubated. After the oral calcium folinate became available, the IV calcium folinate started to be administered at 200 mg once a week. In case number 8, IV calcium folinate treatment was administered at the dose of 25 mg/kg once a month in addition to oral treatment. As this case was resistant to treatment, a ketogenic diet was started, but no benefit was observed from this treatment. Physical therapy was applied to 55.6% of the cases, and vagus nerve stimulation (VNS) was performed on two siblings (22.2%). During the follow-up EEG examinations, VNS was seen to have been useful in bringing the seizures under control.

In general, 88.8% (*n* = 8) of the cases received multiple anticonvulsant treatment, and only case number three, who was diagnosed from family screening only, did not receive anticonvulsant treatment. The drugs used were valproic acid in 77.8% of the cases, clonazepam in 55.6% of cases, topiramate in 22.2% of cases, levetiracetam in 55.6% of cases, lamotrigine in 33.3% of cases, clobazam in 22.2% of cases, rufinamide in 22.2% of cases, lacosamide in 11.1% of cases, and carbamazepine in 11.1% of cases. Before starting the folinic acid treatment, the cases took a mean of 3.2 ± 0.88 anticonvulsant drugs, and this number fell to a mean of 2.8 ± 0.90 after starting folinic acid treatment. As the data for the number of drugs used before and after folinic acid treatment did not conform to normal distribution, these data were examined with the Wilcoxon test and no statistically significant difference was determined (*p* = 0.257).

Of the total cases, 77.8% (*n* = 7) benefitted from the treatment, and there was no effect in 11.1% (*n* = 1), and one case (11.1%) was exitus when treatment was started. Demographic, laboratory and genetic characteristics of the cases and the treatments they received are summarized in Appendix A.

## 4. Discussion

The general clinical, biochemical, and molecular characteristics of cases with FOLR1 mutation are explained in this study. To date, approximately 35 cases from 21 families have been reported in the literature [8,9]. The current study stands out as one of the studies with the highest number of cases reported to date.

As folate is necessary for the synthesis of biogenic amines and pterines in the central nervous system, prenatal and postnatal folate deficiency causes various neurological symptoms such as intellectual disability, epilepsy, ataxia, and pyramidal tract findings [10]. The clinical findings in CFD (irritability and sleep disorders) can start from the age of 4 months [11]. The period from 4.5 months to 4.5 years has been reported in the literature to be an asymptomatic period. The most likely reason for this has been stated to be associated with the intrauterine expression of folate receptor β (FRβ), which provides folate transport in the brain in the early periods of life [4,12]. In the current study, the first complaints of the cases were seen at the age of 3.7 years, while the age at which diagnosis was made was determined to be a mean of 10.4 years. This shows a delay of a mean of 6.7 years.

The first symptoms seen in CFD are motor anomalies such as hypotonia, tremors, ataxia, etc., and delayed speech, followed by developmental regression, and then multi-drug-resistant seizures can emerge [8,12,13,14,15,16]. Other symptoms are irritability, spasticity, autistic behaviour, acquired microcephaly, and polyneuropathy [12,14]. In addition to these findings, “drop” attacks have been reported, which are triggered by certain movements (e.g., washing the face or hands) [8,15]. The most common complaints seen first in the current study cases were determined to be febrile convulsion and attention deficit-hyperactivity-learning difficulty. The physical examination findings observed most often in these cases were ataxia, and neuromotor and intellectual disability (44.4%).

On cranial MRI imaging, delayed myelinization/hypomyelinization and mild cerebral atrophy but more evident cerebellar atrophy have been reported most frequently [8,14]. It has also been reported that bilateral basal ganglia calcifications have been determined on cranial imaging [8,15]. EEG findings of a slow background rhythm and multifocal epileptiform activity have also been reported [14]. Consistent with the literature, white matter involvement was seen in all the current study cases, cerebellar atrophy was prominent in 66.7% of cases and basal ganglia calcification was observed in only one case.

Disruptions in myelinisaton are seen in the early stages of the disease, while cerebral and cerebellar atrophy and T2 hyperintensities in the white matter can be seen on neuroimaging, and low choline/inositol metabolites can be determined on MR spectroscopy [2,12,17,18]. In the current study, a minimal increase in the choline peak and a minimal decrease in the inositol peak were determined in one case.

Papadopoulou et al. published a case report of two siblings. For the first child, treatment was started with multiple anticonvulsant drugs and a ketogenic diet, 6 mg/kg/day oral folinic acid + 10 mg/kg IV calcium folinate twice a week. A decrease in the frequency of seizures and a slow improvement in developmental stages were reported. In the second child, oral folinic acid treatment was started at the same dose, and at the final follow-up examination at the age of 2 years, the developmental stages were reported to be close to those of typically developing peers [12]. In one of the current study cases, ketogenic diet was added to the treatment as the seizures were resistant to both folinic acid and anticonvulsant drugs. No improvement was seen with a ketogenic diet treatment, and ketonemia could not be established in the patient.

The three cases presented by Steinfeld et al. began to exhibit symptoms after the second year of life. Deficiencies in choline and myo-inositol levels were determined on brain MRI spectroscopy. Folinic acid therapy was initiated in two cases after significant clinical and radiological involvement had emerged, and partial clinical improvement was reported. In the third case, treatment was started immediately after movement disorder was seen, and rapid full recovery was observed [4].

In a case report by Kobayashi et al., a fixed gaze lasting several seconds was noticed in a 1.5-year-old patient with no complaints. At the age of 2.5 years, the patient experienced status epilepticus, after which mental regression, progressive ataxia, acquired capability disorder, myoclonic movements, epileptic spasms, and sometimes tonic seizures were observed. Abnormal signals in the white matter were reported on brain MRI, together with abnormal cortical hyperintensity signals (cortical laminar necrosis and ulegyria) in the bilateral temporal lobes [19].

Tabassum et al. reported the case of a patient aged 3.5 years with delayed speech and mild regression in cognitive abilities who presented at the hospital because of tremors and drooling. On brain MRI imaging, abnormality was reported in bilateral white matter intensity, which was seen to be stable over time. The case was determined with the c.665A>G variant in the *FOLR1* gene, and despite EEG findings showing a tendency for seizures, the patient had not experienced a clinical seizure [20].

Congenital microcephaly has been reported in one case in the literature, and acquired microcephaly in five cases [9]. In the current study, there was no congenital microcephaly and acquired microcephaly was determined in 55.6% of the cases.

Normal red blood cell counts and plasma homocysteine and folate concentrations have been reported in patients with FOLR1 deficiency [14]. In the current study, blood folate levels were determined to be close to the lower limit of normal. As the blood folate and metabolite status may not reflect folate deficiency in the central nervous system, the measurement of 5MTHF in spinal fluid is necessary for a diagnosis of CFD [3,11].

CFD is defined as any neurological syndrome associated with low 5MTHF (<5 nmol/L) in CSF, while the serum folate level is in the low normal range [2,10,11,17]. It has been reported that FOLR1 mutations cause a decrease in >80% of the reference value lower limit in the CSF 5MTHF concentration [18]. Due to the limited availability of CSF 5MTHF analysis in many centres across Türkiye and the lack of financial resources for families to obtain this test, CSF 5MTHF levels were not assessed in this study. Therefore, the definitive diagnoses could only be made from genetic analysis and clinical findings.

Grapp et al. examined 72 patients with low 5MTHF concentrations in CSF and determined mutations in the *FOLR1* gene in 10 of 14 patients with a very low 5MTHF level of <45 nmol/L [14]. To date, 28 different mutations in the *FOLR1* gene have been determined; 16 missense/nonsense, 2 splicing, 2 regulatory, 2 small insertions, 1 gross deletion, and 5 complex rearrangements have been found [21]. A total of three different mutations were determined in the current study, including one missense and two nonsense mutations. The variant determined most often was the c.610C>T (p.R204Ter) variant. The c.591C>A (p.Tyr197Ter) variant was determined to be a novel variant not previously reported.

The drugs administered in the treatment of CFD are calcium folinate or folinic acid [11]. The administration of 2–10 mg/kg/day of folinic acid is recommended for CFD due to the FOLR1 mutation [3,9,12,17]. When a sufficient clinical response cannot be obtained with oral treatment, it is recommended that an additional 50–100 mg folinic acid IV is administered once a week alongside intrathecal folinic acid in selected cases [11,12]. Grapp et al. suggested that IV and/or intrathecal folinic acid treatment could be superior to oral treatment [14]. There are different treatment regimens in the literature, such as the combination of 1.7 mg/kg/day PO folinic acid or 8.9 mg/kg/day PO + 500 mg/week IV folinic acid [9]. Folic acid should not be given because of the high binding affinity to FRα and because it could compete with 5MTHF for this receptor [11].

Folinic acid treatment has been shown to provide a return of CSF 5MTHF concentrations to normal and increase glial choline and inositol levels [14]. There has also been reported to be a decrease in the frequency of epileptic seizures and an improvement in motor skills within 2 months of treatment [14]. In two cases reported by Potic et al., an improvement was seen in myelinisaton on brain MRIs after starting treatment [9]. While improvements have also been seen in children who started folinic acid treatment before the age of 6 years, it has been reported that there is less benefit when started later [11].

Delmelle et al. reported that two siblings with developmental delay, seizures, and ataxia were started on treatment of 2 mg/kg/day folinic acid, and the dose was then increased to 5 mg/kg/day. When a sufficient response was not obtained to this treatment, 6 mg/kg/day IV folinic acid was given in four doses throughout 24 h followed by 12 mg/kg/day in four doses throughout 48 h. An extremely dramatic positive response was reported to be obtained with this. Then, both cases continued with 7 mg/kg/day oral folinic acid and 20–25 mg/kg IV folinic acid once every 4 weeks (in four doses over 24 h). Extremely good clinical improvement was obtained with this treatment in the younger child, and although the seizures continued in the older child, there was reported to be a great decrease in the frequency of seizures compared to before treatment [2]. In another study, two siblings were diagnosed with CFD at the ages of 33 and 28 years. With folinic acid treatment, there was reported to be a decrease in seizure frequency and in the anticonvulsant drug dose [22]. The cases in the current study received a mean dose of 5.2 mg/kg/day oral folinic acid treatment. In addition, most cases received 100 mg folinic acid IV once a week, one case received 200 mg/week, and another received 25 mg/kg once a month. The vast majority of the cases in this study were seen to have great clinical benefit from the treatment.

Based on our study, it can be concluded that *FOLR1*-related CFD should be considered in cases characterized by febrile convulsion, the delayed acquisition of developmental milestones, ataxia, autism spectrum disorder, acquired microcephaly, and the predominant findings of white matter involvement and cerebellar atrophy on brain MRI. Since the disease has an asymptomatic period in early childhood, the diagnosis of CFD may be delayed. Treatment initiated after the onset of symptoms, particularly after the age of six, may not yield a sufficient response. Therefore, we believe that incorporating *FOLR1* gene analysis into newborn screening programmes could be highly beneficial for early diagnosis and timely treatment.

An experimental rat model study reported that diet-related folate deficiency reduced the amount of phosphatidylcholine in the brain membrane, and this could be prevented with L-methionine supplementation [18]. One of the first publications on this subject in the literature reported the combination of FOLR1-related CFD and LAMM syndrome. Neurological abnormalities were noticed when the case was 6 months old. As seizures could not be brought under control with folinic acid and anticonvulsant treatment, pyridoxal 5 phosphate was added to the treatment, and subsequently, the seizures could be brought under control [23].

It has been reported in the literature that there could be an anti-folate effect of some anticonvulsant drugs such as valproate, phenobarbital, primidone, phenytoin, carbamazepine, oxcarbazepine, topiramate, gabapentine, and pregabaline but no such effect has been reported for anticonvulsants such as lamotrigine, levetiracetam, clobazam, and clonazepam [9]. The number and doses of anticonvulsant drugs used by the current study patients before folinic acid treatment were evaluated, and it was determined that there was a decrease in the number and doses of these drugs following treatment. In addition, a VNS device was applied to cases in this study and on follow-up EEGs, and this was seen to have been effective in reducing epileptic activity.

## 5. Conclusions

This study presented nine cases diagnosed with CFD who were followed up at two different centres in Türkiye. This study can be considered of value as one of the largest case series in the literature on this subject. A novel variant of the *FOLR1* gene mutation not previously reported was determined as a result of the study analyses. The application of VNS in addition to folinic acid and the anticonvulsant drug treatment seemed to be effective in seizure control. Nevertheless, there is a need for further comprehensive studies with larger patient series to be able to discover more effective curative treatments.

## Figures and Tables

**Figure 1 diagnostics-15-00892-f001:**
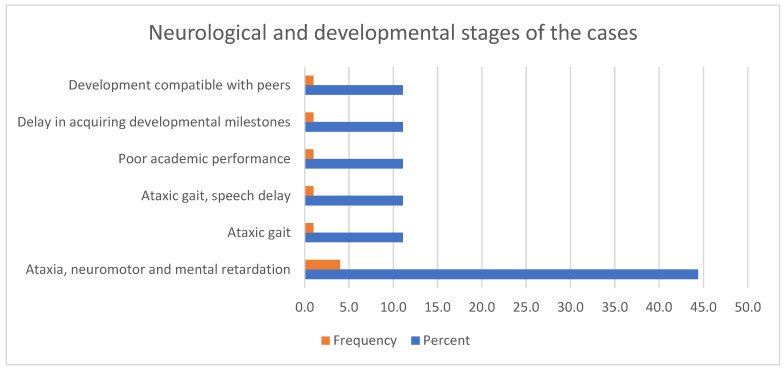
The neurological and developmental stages of the cases are illustrated.

**Table 1 diagnostics-15-00892-t001:** The first provisional diagnoses received by the cases before the definitive diagnosis.

	Frequency	Percent
Epilepsy	4	44.4
Developmental delay	2	22.2
Cerebral palsy	1	11.1
Hiperactivity	1	11.1
Family screening	1	11.1

**Table 2 diagnostics-15-00892-t002:** The first complaints noticed during the diagnosis process of the cases.

Initial Application Complaints	Frequency	Percent
Febrile convulsion	2	22.2
Developmental delay	1	11.1
Powerlessness	1	11.1
Hyperactivity, learning disability	1	11.1
Speech delay	2	22.2
autism spectrum disorder	1	11.1
Difficulty walking	1	11.1

**Table 3 diagnostics-15-00892-t003:** The complete blood count and biochemical results of the cases are shown.

	Minimum	Maximum	Mean	Std. Deviation
Serum Folat (ng/mL)	1.3	4.0	3.1	0.80
Total Homosistein (µmol/L)	3.6	26.9	13.1	8.33
Hemoglobin (g/dL)	9.7	13.9	12.2	1.36
MCV (fL)	76.2	94.1	82.9	5.48

## Data Availability

Data are available on request due to restrictions (e.g., privacy, legal or ethical reasons).

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
