# Peer review of "A Clinical and Genetic Evaluation of Cases with Folate Receptor α Gene Mutation: A Case Series from Türkiye"

_diagnostics, 2025, doi:10.3390/diagnostics15070892_

Round 1
Reviewer 1 Report
Comments and Suggestions for Authors
The submitted paper (A CLINICAL AND GENETIC EVALUATION OF CASES WITH FOLR1 GENE MUTATION: A CASE SERIES FROM TÜRKİYE ) has a thorough clinical and genetic analysis of 9 individuals with FOLR1 gene mutations presenting with cerebral folate transporter deficiency (CFD) from 2 tertiary centers in Turkiye. The investigation by the authors highlights mean delay of diagnosis by 6.7 years from symptom initiation and calls out febrile convulsions and ADH-learning problems as frequent initial manifestations. It details three FOLR1 variants, including this new c.591C>A (p.Y197Ter) mutation. High rate of parental consanguinity (88.9%) implies strong genetic background. MRI evidence of white matter involvement, cerebellar atrophy, and cerebral atrophy concurs with the previous literature. Oral folinic acid (mean 5.2 mg/kg/day) and intravenous folinate treatment was effective clinically in most patients, although one patient died of the disease. Vagus nerve stimulation (VNS) applied in two cases was also helpful for seizure control. In spite of the retrospective nature of the study and lack of CSF 5MTHF levels (can these be done and included??), it is rich in information regarding diagnosis, treatment outcomes, and genetic differences in CFD. The results add a lot to the literature but call for larger, prospective studies to further optimize therapeutic approaches. The role of VNS as an adjunctive therapy needs to be investigated further. The paper may be accepted in its current form.
Author Response
Comments 1: The submitted paper (A CLINICAL AND GENETIC EVALUATION OF CASES WITH FOLR1 GENE MUTATION: A CASE SERIES FROM TÜRKİYE ) has a thorough clinical and genetic analysis of 9 individuals with FOLR1 gene mutations presenting with cerebral folate transporter deficiency (CFD) from 2 tertiary centers in Turkiye. The investigation by the authors highlights mean delay of diagnosis by 6.7 years from symptom initiation and calls out febrile convulsions and ADH-learning problems as frequent initial manifestations. It details three FOLR1 variants, including this new c.591C>A (p.Y197Ter) mutation. High rate of parental consanguinity (88.9%) implies strong genetic background. MRI evidence of white matter involvement, cerebellar atrophy, and cerebral atrophy concurs with the previous literature. Oral folinic acid (mean 5.2 mg/kg/day) and intravenous folinate treatment was effective clinically in most patients, although one patient died of the disease. Vagus nerve stimulation (VNS) applied in two cases was also helpful for seizure control. In spite of the retrospective nature of the study and lack of CSF 5MTHF levels (can these be done and included??), it is rich in information regarding diagnosis, treatment outcomes, and genetic differences in CFD. The results add a lot to the literature but call for larger, prospective studies to further optimize therapeutic approaches. The role of VNS as an adjunctive therapy needs to be investigated further. The paper may be accepted in its current form.
Response 1: I sincerely appreciate the time and effort you dedicated to reviewing my manuscript. Thank you also for recommending its publication in Diagnostics. I am honored by your positive assessment and hope the final version meets the journal’s standards.
Reviewer 2 Report
Comments and Suggestions for Authors
Dear Editors,
Thank you for the opportunity to review this manuscript on Cerebral Folate Transporter Deficiency (CFTD). This rare autosomal recessive disorder, typically caused by mutations in the FOLR1 gene encoding the folate receptor alpha (FRα) protein. With 28 cases, this series stands as one of the largest in the literature and notably identifies a previously unreported novel variant, making it both a significant and valuable contribution to the field.
I have only one minor suggestion: Given the rarity of CFTD, I would recommend that the authors elaborate on how their findings could improve the diagnosis and treatment of this condition. These clinical implications should be summarized and highlighted in both the abstract and discussion sections to emphasize the important translational value of this research.
Author Response
Comments 1: Thank you for the opportunity to review this manuscript on Cerebral Folate Transporter Deficiency (CFTD). This rare autosomal recessive disorder, typically caused by mutations in the FOLR1 gene encoding the folate receptor alpha (FRα) protein. With 28 cases, this series stands as one of the largest in the literature and notably identifies a previously unreported novel variant, making it both a significant and valuable contribution to the field.
I have only one minor suggestion: Given the rarity of CFTD, I would recommend that the authors elaborate on how their findings could improve the diagnosis and treatment of this condition. These clinical implications should be summarized and highlighted in both the abstract and discussion sections to emphasize the important translational value of this research.
Response 1:
I sincerely appreciate the time and effort you dedicated to reviewing my manuscript.
In line with your suggestions, the following text has been incorporated into the Discussion section.
“Based on our study, it can be concluded that FOLR1-related CFD should be considered in cases characterized by febrile convulsion, delayed acquisition of developmental milestones, ataxia, autism spectrum disorder, acquired microcephaly, and predominant findings of white matter involvement and cerebellar atrophy on brain MRI. Since the disease has an asymptomatic period in early childhood, the diagnosis of CFD may be delayed. Treatment initiated after the onset of symptoms, particularly after the age of six, may not yield sufficient response. Therefore, we believe that incorporating FOLR1 gene analysis into newborn screening programs could be highly beneficial for early diagnosis and timely treatment.” See page 8, paragraph 3.
Similarly, the text below has been added to the Abstract.
“Our study suggests that FOLR1-related CFD should be considered in cases with febrile convulsions, developmental delay, ataxia, autism spectrum disorder, acquired microcephaly, and MRI findings of white matter involvement and cerebellar atrophy. Due to an asymptomatic early period, CFD diagnosis may be delayed, and treatment after symptom onset may be less effective. Incorporating FOLR1 gene analysis into newborn screening programs could facilitate early diagnosis and treatment.” See Abstract on page 1.